# OpenReview forum: "A Behavioural and Representational Evaluation of Goal-Directedness in Language Model Agents"
_ICML.cc/2026/Conference — ICML 2026 regular_

### Official Review · Reviewer_ajN5 · 2026-03-09

**Soundness:** 3
**Presentation:** 3
**Significance:** 3
**Originality:** 3
**Overall Recommendation:** 5
**Confidence:** 3

**Summary:**

The paper proposes a method for evaluating the goal-directedness of language agents through a combination of behavioural tests and comparing these with evaluations on the internal world models (cognitive maps) extracted with supervised probing during inference. This makes progress towards solving a fundamental issue with previous goal-directedness evals, where it is generally impossible to disentangle failures of agency (intention, planning) from failures of competence and robustness purely through behavioural testing. The authors attempt to extract the current world state from the agent, and use this to predict the next move made by the agent, assuming that it is a performing optimal planning with respect to its internal representation. This is then compared to the optimal moved made with the ground truth board state. If the world model is better at predicting the agents moves (or errors), it is strong evidence that failures are due to goal-directed planning on a faulty world model, rather than a lack of internal planning at all. The authors also observe a representation shift during the reasoning process, heralding a change from world-model-like representations to narrow focus on next action selection, which further evidences that internal planning and hence goal-directness is going on in inside these models.

**Compliance With Llm Reviewing Policy:**

Affirmed.

**Final Justification:**

The authors have made a lot of positive changes to the paper, and it looks like they have removed the main weakness (the subjective model perhaps not being a better explanation of the agent's behaviour than the ground truth).

**Key Questions For Authors:**

Questions

1. Can the authors be sure that the 4 layer transformer decoder is not expressive enough to `cheat by continuation heuristics’ in the one-shot setting they test it in? It seems plausible that these models, when inputted with the full world state, have the capacity to perform the navigation algorithm themselves. And it wouldnt be too hard to check. Without controlling for this confounder, its hard to trust the results. Or, the authors could use a less expressive model.

2. What about the reverse recovery statistic, where the agent takes a move that is suboptimal in the cognitive map, but is optimal in the ground truth?

3. To check for a linear representation of the board state, would it not be better to train a linear probe to classify every square on the grid independently, rather than one linear probe to predict the type for any square, conditional on (x, y)? A separate linear probe for each coordinate can learn exactly which neurons represent that specific square, whereas a single linear probe for all (x, y) is forced to use the exact same neurons to guess every single square on the board. This could plausibly explain why linear probing performed poorly.

4. Also, they could try other choices of board representation that may be closer to the agents true cognitive map. for example in [1] they realised the model didnt didn't linearly encode the board in absolute terms ("black piece" vs "white piece"), but rather in relative, state-dependent terms. Did the authors try any other board state encodings, e.g. relative to the agent’s position? Or affordances (like legal v.s. illegal moves)?

5. Why did you choose not to do an interventional study to determine if the cognitive map was driving the agent's behaviour?

[1] Nanda, Neel, Andrew Lee, and Martin Wattenberg. "Emergent linear representations in world models of self-supervised sequence models." Proceedings of the 6th BlackboxNLP Workshop: Analyzing and Interpreting Neural Networks for NLP. 2023.

**Limitations:**

yes

**Strengths And Weaknesses:**

Strengths

1. The paper is well written, and deals with a timely and difficult problem. The results are interesting.

2. Section 5.2 where they calculate the agents actions relative to a decoded internal cognitive map is a great idea. This is basically taking a step towards a subjective expected utility model of the agent, which lets you take into account failures in the agent’s `beliefs’ when predicting their behaviour.  If this could be convincinly nailed down, it would be a really strong result!

3. These results are nicely tee’d up by experiments with distractor variables, which add to the overall picture

Weaknesses

1. As noted by the authors, it is `somewhat surprising’ that the action accurate relative to the decoded cognitive map is lower than the accurate relative to the ground truth. This makes it hard to conclude that they have found a better representation of the agent’s beliefs than just the ground truth world state, which should be more predictive than the input. Without this, its hard to draw the main conclusions.

2. Unlike [1], they dont try to establish that the representations of the world state are actually used by the agent, e.g. intervening on them (e.g. activation patching) and observing the expected downstream change in policy. Its not clear why this is omitted, but it woudl greatly improve how convincing the result is.

3. There are some potential issues with the linear probing scheme (not used in the main result), and with how they construct the cognitive map from the probe output. It feels like the results could be greatly improved with some more tinkering here. see questions below.

[1] Li, Kenneth, et al. "Emergent world representations: Exploring a sequence model trained on a synthetic task." arXiv preprint arXiv:2210.13382 (2022).

---

> ### Author Rebuttal · Authors · 2026-03-31
>
> We would like to thank Reviewer ajN5 for their careful engagement with our work. Below, we address each of their concerns.
>
> **Decoded cognitive map predicting actions worse than ground truth**
>
> Following this comment, we revisited the action accuracy analysis, using top-k decoded positions for the agent and goal, rather than the single highest-probability decoding (top-1) used in our submission. For k ∈ {3, 5, 10}, we computed an upper envelope over the k×k possible decoded grids (excluding agent-goal collisions) and a probability-weighted variant. [This figure](https://ibb.co/hFvVTdYx) shows our results. At higher grid sizes and complexities, we found that the agent’s actions are closer to optimal, with respect to the top k envelope than they are with respect to the ground truth. Moreover, the effect is even more pronounced for increased grid sizes than for increased complexity. We interpret this as evidence that top-1 decoding collapses the agent’s uncertainty, and that instead, as the state space grows and the agent’s representations become fuzzier, actions are better characterised as planning under a distribution over plausible states.
>
> **Q1: Could the 4-layer decoder cheat by performing navigation itself?**
>
> To address this, we trained a 2-layer decoder on the same task. Pre-reasoning, it achieved 41.5% first-step accuracy (comparable to the 4-layer decoder), while post-reasoning it reached 66.5%, surpassing the four-layer model (~55%). We interpret the stronger generalisation of the less expressive decoder as evidence against the hypothesis that the decoder succeeds by internally performing navigation. For the final version of the paper, we will extend this analysis to even weaker decoders that are increasingly unlikely to implement A*.
>
> **Q2: Reverse recovery statistic (agent suboptimal in cognitive map, optimal in ground truth)?**
>
> We calculated the suggested reverse recovery statistic _using the top-1 decoding of the original paper_.
>
> | Grid Size | Reverse Recovery |
> | :--- | :--- |
> | 7 | 86.6% |
> | 9 | 84.6% |
> | 11 | 64.9% |
> | 13 | 67.4% |
> | 15 | 61.1% |
>
> | Density Value | Reverse Recovery |
> | :--- | :--- |
> | 0.0 | 100.0% |
> | 0.2 | 67.6% |
> | 0.4 | 63.5% |
> | 0.6 | 68.1% |
> | 0.8 | 69.3% |
> | 1.0 | 60.9% |
>
> We observe that reverse recovery declines as grid size and complexity increase. We interpret this as reflecting growing uncertainty in the agent’s internal representation of the world state as a function of environment complexity.
>
> **Q3: Per-square linear probes**
>
> Our setup with position-agnostic linear and MLP probes was motivated by initial experiments with position-specific linear probes, like those described in the reviewer's question, which proved ineffective at identifying cell types across all grid sizes. While we did not conduct comprehensive ablations on probe design, we note that a single MLP can currently be trained across all grid sizes with padding cells and applied to produce cognitive maps with satisfactory performance.
>
> **Q4: Alternative grid state encodings (relative, affordance-based)**
>
> We thank the reviewer for the insightful observation. We agree that probing alternative encodings of the grid could reveal interesting ways the model represents affordances. In Appendix F.3, we experimented with Goal distance probes to predict the current A* distance between the agent and goal position, finding that the correct distance (which could span more than 50 steps for larger grids) could be reliably predicted with a MAE of ~3, using activations from positions before and after reasoning. We agree that additional experiments along these lines would help narrow down usable encodings of the environment state in model representations, and should be a priority for follow-up studies.
>
> **Q5: Why no interventional study?**
>
> We did not perform this initially because causal mediation is not a primary point of inquiry in our work, but we fully agree that interventional studies would complement the cognitive map results. We have now performed activation patching on the residual stream of GPT-OSS-20B, both at individual layers and across all layers, using corrupted examples that move either the agent or the goal. We find that patching changes the model’s action only when applied across all layers, and only at the grid state tokens or the pre-action token.
>
> We take this to suggest that grid, goal, and agent position information is distributed across layers rather than localised to a single one. This aligns with Choi et al. (2025), who likewise find that reading is often possible from a single middle layer, while reliable steering requires intervention across all layers. In our case, cognitive maps are readable with probes across individual layers, but causal intervention appears to require broader manipulation than the probes used here. We will include these quantitative results in the camera-ready version of the paper.
>
>
> ----
> Choi et al. (2025). Scalably Extracting Latent Representations of Users.

---

> > ### Author Rebuttal · Reviewer_ajN5 · 2026-04-02
> >
> > The authors have answered all my questions and so I am increasing my score.
> >
> > The new plots showing the `uncertain' world model is more predictive than the ground truth strengthen the result a lot, but would do so even more if there were some error bars / statistical significance establishing that the subjective world model is a better explanation of the agent's behaviour, rather than eyeballing it.

---

### Official Review · Reviewer_CJd1 · 2026-03-13

**Soundness:** 2
**Presentation:** 2
**Significance:** 2
**Originality:** 2
**Overall Recommendation:** 3
**Confidence:** 5

**Summary:**

The paper proposes a white-box framework to evaluate goal-directed behaviour in LLM-based agents in a grid-world navigation task under varying levels of difficulty, environmental perturbations, and multi-goal settings.by combining behavioural testing with probing of internal representations.Their analysis reveals that the model encodes interpretable cognitive maps and multi-step action plans, with representations shifting during reasoning from broader planning to immediate action selection.

**Compliance With Llm Reviewing Policy:**

Affirmed.

**Final Justification:**

I appreciate the authors' thorough response and new experiments. While the methodological contribution is clearer, concerns about generalizability remain. Adjusting my score accordingly.

**Key Questions For Authors:**

1. Are the claims about general agents, or only about LLM behaviour in structured tasks?
2. How would the conclusions change in partially observable or stochastic environments?
3. To what extent do findings generalize to real-world agent settings?

**Limitations:**

yes

**Strengths And Weaknesses:**

Strengths:
1. The authors propose an evaluation framework using both behaviour and internal representations, which is currently important to understand goal-directedness + LLM interpretability + safety.
2. They use controlled grid-world experiments and iso-difficulty transformations to analyse robustness, bias, and instrumental goal handling systematically.
3. They introduce interpretability analysis (cognitive maps, plan decoding, recovery metric) that provide insight into belief-action consistency and planning representations.

Weaknesses:

1. Fully observable grid worlds are very simplified. May not reflect real-world agent reasoning, especially where uncertainty, memory, and exploration are critical.

2. Goal-directedness is studied only in navigation, which may not capture broader aspects like planning under uncertainty, long-horizon reasoning.

3. Excluding partially observable settings limits claims about goal-directedness in realistic environments. For partially observable settings, belief update is crucial.

4. Single model evaluation on GPT-OSS-20B. It raises concerns about generalizability across architectures/sizes.

5. The observation that action accuracy with respect to the decoded cognitive map is consistently lower than that with respect to the ground truth is mentioned as surprising but not thoroughly analyzed.

---

> ### Author Rebuttal · Authors · 2026-03-31
>
> We would like to thank Reviewer CJd1 for their review. Below, we address their concerns on the scope and generalisability of our results.
>
> **Fully observable grid worlds are simplified and might not reflect real-world settings**
>
> Our choice for a manageable, fully observable setup was deliberate and motivated by several factors. First, the intended contribution of our paper is primarily methodological, and our evaluation is intended primarily to provide a blueprint for combining behavioral and representational approaches in the study of agent behavior. Additionally, we would like to point the reviewer to the work of other authors, such as Li et al. (2023), who have employed a similar methodology, using the fully observable game of Othello to study environment representations in LMs. The result by Li et al. (2023) proved influential precisely because their controlled settings make their findings straightforward to interpret.
>
> Similarly, environment variables in the gridworld navigation task can be precisely controlled, which enabled robustness tests such as iso-difficulty transforms and instrumental goals. Moreover, this setup provided us with a limited action set, a concise environment representation, and simple update rules, which allowed us to quickly scale our representational analyses to the reasoning tokens of a 20B LM. We agree that understanding how these evaluations could scale to more complex tasks and scenarios, with their inevitable confounding factors, is an important question and we intend to study it in future work.
>
> **Excluding partially observable settings limits claims about goal-directedness**
>
> In Appendix A.4, we report preliminary results showing that GPT-OSS-20B performs poorly in partially observable navigation settings. This led us to defer goal-directedness evaluations in such settings to future work with more capable frontier systems. As discussed in §3 (lines 151-164), partially observable environments introduce a range of additional questions that fall beyond the scope of this paper. For example, our fully observable setup admits a well-defined optimal policy, enabling analyses of behaviour optimality that are not feasible under partial observability.
>
> **Evaluation is limited to a single model**
>
> At the time of our experiments, GPT-OSS-20B was the best open-source reasoning model sufficiently small to enable representational analyses on our computational infrastructure. We also ran preliminary behavioural tests with Qwen3-30B-A3B-Thinking-2507, but excluded it due to unreliable performance even in the fully observable setting. These constraints (i.e., strong capability at small enough size) limited the set of viable agents; we partly mitigated this by sampling multiple trajectories per environment. That said, our methodological contribution is designed to be model-agnostic, and we intend to apply it to additional models, including larger and more recent reasoning models, to strengthen generalisability.
>
> **Decoded map action accuracy is consistently lower than ground truth action accuracy**
>
> We thank the reviewer for this comment, which prompted us to revisit our analysis in §5.2. While the current results in Table 2 employ cognitive maps in which each cell is decoded as the argmax over the available cell types by the probe, we relax this assumption by considering the optimality of an action given a set of top-k agent and goal positions per grid. This mitigates the low precision of cognitive map probes for goal and agent location prediction. For k ∈ {3, 5, 10}, we measure action optimality and plot the proportion of optimal actions for Top-1 decoded grid (previous results), Top-k positions, and include a Top-k variant where the probability of agent and goal decoding is used to weight action optimality. [This figure](https://ibb.co/hFvVTdYx) shows our results.
>
> We observe that Top-k accuracy results are consistently higher than accuracy against the ground truth grid across grid sizes and complexity bins. These results suggest that the reconstruction obtained by argmax over cognitive map probe predictions is lossy and does not capture the full extent of environment representations. By accounting for multiple viable options based on probe predictions, we find that decoded grids can indeed explain a higher proportion of actions than ground-truth ones.
>
> **Q1: Are the claims about general agents, or only about LLM behaviour in structured tasks?**
>
> Addressed above. Note our core claim that behavioural evaluation is not sufficient applies to general agents, and even more so for realistic tasks where the assumptions of classic behavioural methods are untenable.
>
> **Q2: How would the conclusions change in partially observable or stochastic environments?**
>
> Addressed above, and by appendix A.
>
> **Q3: To what extent do findings generalise to real-world agent settings?**
>
> Addressed above.
>
> ---
> Li et al. (2023) Emergent world representations: Exploring a sequence model trained on a synthetic task.

---

> > ### Author Rebuttal · Reviewer_CJd1 · 2026-04-04
> >
> > I thank the authors for their thorough response. The methodological framing and the Li et al. (2023) analogy partially address my concerns, and the new top-k analysis strengthens the results. However, the single-model, fully-observable, navigation-only evaluation still limits the generalizability of the claims. I am raising my score from 2 to 3, though I maintain that broader validation is needed.

---

### Official Review · Reviewer_pRis · 2026-03-13

**Soundness:** 4
**Presentation:** 3
**Significance:** 4
**Originality:** 4
**Overall Recommendation:** 5
**Confidence:** 4

**Summary:**

This paper studied goal-directedness in LLM through behavioral and representational evaluation. They focus on grid world problems with different difficulties. In the behavioral study, they evaluated how well did the LLM (gpt-oss-20b) play the game from different perspectives. Then they connect behavioral study to representational study, interpreting different aspects of goal-directedness, such as internal world modeling, planning, etc. This study reveals that LLM agents internally models the grid world and the internals are decodable and are roughly aligned with the explicit / behavioral decisions.

**Compliance With Llm Reviewing Policy:**

Affirmed.

**Key Questions For Authors:**

1. Is it possible to make this more actionable, for example to steer the model internals?
2. Does LLM training matter? If we spend more effort on training the LLM to understand the world and on making better decisions, will the results of representational study change? My intuition is that achieving high behavioral scores doesn't guarantee that good representations are learned.

**Limitations:**

yes

**Strengths And Weaknesses:**

Strengths:
1. The motivation is strong. The paper is dense, compact and easy to follow.
2. The experiments are comprehensive, clearly connect behavioral and representational studies, making a strong evidence that studies can be and should be conducted from both perspectives simultaneously.

Weaknesses:
1. Minors: figures can be improved. For example, legends are too small in Fig. 4.

---

> ### Author Rebuttal · Authors · 2026-03-31
>
> We are very grateful to Reviewer pRis for their positive assessment of our work.
>
> **Q1: Is it possible to make this more actionable, for example to steer the model internals?**
>
> In response to this question, we conducted some preliminary activation patching experiments. We found that patching across all layers for all prompt tokens corresponding to grid cells was effective at changing the agent’s action output, as was intervening on the token immediately preceding the action output in flipping the prediction. However, single-layer interventions applied to the chat  template tokens used in our previous probing experiments proved ineffective, which suggests an interesting asymmetry between readouts and interventions. We note that these results are in line with recent work showing that prompt-related attributes can be read by probing classifiers at various points in the forward pass, but interventions require more precise localisation to achieve a measurable impact (Choi et al., 2025). Identifying whether specific layers and positions enable intervention strategies for our gridworld setup is an interesting direction we would like to explore in future work.
>
> **Q2: Does LLM training matter?**
>
> We agree with the reviewer that this is an important open question, and that high behavioural performance need not guarantee that a model has good internal representations. In fact, detecting such a gap is the key motivation behind our approach, combining behavioural and representational analyses. In our setup, we employ an existing LLM without additional fine-tuning, but we agree that studying how the relationship between the agent’s behavioural performance and its internal representational quality varies across training stages and model scales is an important future step. Our work introduces a framework that would naturally support such analyses at scale, and provides initial evidence of its relevance for complementing behavioural analyses.
>
> **Minors: figures can be improved. For example, legends are too small in Fig. 4.**
>
> We will polish and improve the readability of figures for the final version of the paper.
>
>
> -----
> Choi et al. (2025). Scalably Extracting Latent Representations of Users.

---

> > ### Author Rebuttal · Reviewer_pRis · 2026-04-03
> >
> > I have no further concerns so I'm keeping my positive score as is.

---

### Decision · Program_Chairs · 2026-04-30

**Decision:**

Accept (regular)

**Comment:**

The paper received two accepts (pRis, ajN5) and one weak reject (CJd1). Reviewer ajN5 raised their soundness score from 2 to 3 after the rebuttal and confirmed accept in the final discussion. The rebuttal introduced a top-k cognitive map analysis showing that the agent's uncertain world model is more predictive of its actions than the ground truth at larger grid sizes, and preliminary activation patching experiments. All reviewers found these additions substantively strengthened the paper.

This paper makes a solid contribution. The framework of combining behavioral evaluation with representational analysis to characterize goal-directedness is well-motivated and methodologically sound.

 The primary concern, shared across reviews, is that the evaluation is simple: a single environment type (grid world), a single task (navigation), full observability, and a single model. This is fair, but I view the paper as a good first step: the contribution is the framework and methodology, not a definitive claim about goal-directedness in the wild.